# Genome-Wide Identification of lncRNAs During Rice Seed Development

**DOI:** 10.3390/genes11030243

**Published:** 2020-02-26

**Authors:** Juan Zhao, Abolore Adijat Ajadi, Yifeng Wang, Xiaohong Tong, Huimei Wang, Liqun Tang, Zhiyong Li, Yazhou Shu, Xixi Liu, Shufan Li, Shuang Wang, Wanning Liu, Jian Zhang

**Affiliations:** State Key Lab of Rice Biology, China National Rice Research Institute, Hangzhou 311400, China; zhaojuan521321@163.com (J.Z.); threetriplea@yahoo.com (A.A.A.); wangyifeng@caas.cn (Y.W.); tongxiaohong@caas.cn (X.T.); wangyingkai2006@126.com (H.W.); liquntang2013@126.com (L.T.); lzhy1418@163.com (Z.L.); mm123456m@126.com (Y.S.); 18338690086@163.com (X.L.); 13126890093@163.com (S.L.); a778211546@163.com (S.W.); dearliuwanning@126.com (W.L.)

**Keywords:** Rice (*Oryza sativa* L.), lncRNAs, seed development, grain size

## Abstract

Rice seed is a pivotal reproductive organ that directly determines yield and quality. Long non-coding RNAs (lncRNAs) have been recognized as key regulators in plant development, but the roles of lncRNAs in rice seed development remain unclear. In this study, we performed a paired-end RNA sequencing in samples of rice pistils and seeds at three and seven days after pollination (DAP) respectively. A total of 540 lncRNAs were obtained, among which 482 lncRNAs had significantly different expression patterns during seed development. Results from semi-qPCR conducted on 15 randomly selected differentially expressed lncRNAs suggested high reliability of the transcriptomic data. RNA interference of *TCONS_00023703*, which is predominantly transcribed in developing seeds, significantly reduced grain length and thousand-grain weight. These results expanded the dataset of lncRNA in rice and enhanced our understanding of the biological functions of lncRNAs in rice seed development

## 1. Introduction

Long non-coding RNA (lncRNA) refers to transcripts longer than 200 nucleotides and are functional RNA molecules which have no discernable coding potential [1,2]. In eukaryotes, lncRNA could be transcribed from either genic or intergenic regions of the genome by RNA polymerase II and III and shares many common features with messenger RNAs (mRNAs), including 5′ capping, splicing and polyadenylation [3]. lncRNAs have been classified into three categories based on their genomic locations and in relation with the protein-coding genes. These include lincRNAs (long intergenic non-coding RNAs), intronic non-coding RNAs and lncNATs (long nature antisense transcripts) [4]. Previously thought of as meaningless transcriptional noise, lncRNA has now been fully recognized as an important regulator in diverse biological processes across eukaryotes such as cell cycle regulation, cellular growth and differentiation [5]. In humans, disorders of lncRNA expression have been associated with cancer and used as molecular markers in medical diagnosing. *HOTAIR* encoding an intergenic lncRNA was a well-documented example which was implicated in breast cancer development [6]. Various regulatory mechanisms have been proposed for the lncRNA function, including epigenetic modulation, transcriptional regulation, promoter occlusion, genomic imprinting, alternative splicing, subcellular transport and as a decoy to catch proteins or microRNAs [4]. 

Given its critical roles in gene regulation, lncRNAs have received much attention in the science community. For plants, a large number of lncRNAs have also been identified and proved to play crucial roles in gene silencing, reproductive development, flowering time regulation, stress responses and other important developmental pathways [7]. The famous antisense lncRNAs *COOLAIR* (cold-induced long antisense intragenic RNAs) was considered to function in vernalization of *Arabidopsis* through transcriptional interference [8]. In rice, the Long-Day Specific Male-fertility Associated RNA (*LDMAR*) has been proved to be required for normal pollen development under long-day conditions [9]. Additionally, more lncRNAs have been brought to light in plants. However, only a few lncRNAs have been clarified with a clear regulatory mechanism at present.

Systematic identification of lncRNAs at the genome-wide level is certainly crucial to understanding their biological functions. Wang et al. [10], using the Reproducibility-based Tiling-array Analysis Strategy (RepTAS) and strand-specific RNA sequencing (RNA-seq), predicted 37,238 lncNATs in *Arabidopsis* and investigated their expression in response to light. Using the directional and non-directional high-throughput RNA-seq experiments, 27,065 and 22,814 transcripts were identified in rice and maize, respectively [11]. Strand-specific RNA-seq also revealed 2224 lncRNAs during rice reproduction [12]. 

Rice (*Oryza sativa* L.) is a major food crop and model plant for biological research. As a reproductive organ of rice, studying seed growth and development is of great significance. In this study, we performed the paired-end RNA-seq with samples obtained from Nipponbare pistils, seeds after flowering 3 and 7 days. We systematically identified 540 rice lncRNAs, among which 482 lncRNAs showed differential expression during seed development. The stage-specific expression patterns of 15 lncRNAs were verified by semi-qRT-PCR. Moreover, an RNA interference (RNAi) mutant of a seed-specific lncRNA *TCONS_00023703* showed significantly reduced grain length and thousand-grain weight, indicating that lncRNAs participate in the regulation of rice seed development.

## 2. Materials and Methods

### 2.1. Plant Materials

All plant samples used in this study were of the rice variety *Oryza sativa* L. ssp. *Japonica* cultivar Nipponbare (NIP). Plants were planted in a paddy field in Hangzhou, Zhejiang Province under normal conditions. For developing seed collection, each panicle was labelled on the anthesis day. Then, mature pistils before pollination (termed as 0 days after pollination or DAP) and seeds after pollination at 3 and 7 days (termed as 3 and 7 DAP) were collected manually with glumes removed and frozen immediately in liquid nitrogen for further analysis. Callus was induced from mature seeds according to previous reports [13].

### 2.2. Paired-End RNA Sequencing

Seeds at 0, 3, 7 DAP were used for RNA-seq. Nanodrop (Thermo Fisher, Shanghai, China), Qubit 2.0 (Invitrogen, Carlsbad, CA, USA), and Agilent BioAnalyzer 2100 (Beijing, China) were used to detect the purity, concentration, and integrity of RNA samples to ensure that transcriptome sequencing was performed using qualified samples. Complementary DNA (cDNA) library construction and paired-end sequencing were performed by Biomarker Technologies (Beijing, China). The cDNA library concentration and insert size were detected using Qubit 2.0 and Agilent BioAnalyzer 2100, respectively. The effective concentration of the library was accurately quantified using qPCR (Kapa Biosystems, Woburn, MA, USA) to ensure the library quality. The resulting library was sequenced using an Illumina Hiseq 2500 device (Illumina, San Diego, CA, USA) that generated the paired-end reads of 100 nucleotides.

### 2.3. The Long Non-Coding RNAs Prediction

The RNA-seq datasets were aligned to rice genome IRGSP-1.0 with TopHat2 to obtain mapped reads [14]. Mapped reads were assembled to transcripts using Cufflinks and Scripture [15]. After discarding known mRNAs and transcripts FPKM < 0.5 (fragments per kilobase of transcript per million fragments mapped) with potential coding capacity, the non-coding transcripts longer than 200 bp with FPKM score ≥ 1.5 were identified as final rice lncRNAs from the samples. The potential coding capacity of transcripts were screened through CPC (coding potential calculator), CNCI (coding-non-coding index) and Pfam analysis (https://pfam.xfam.org/). If CPC score < 0, CNCI score < 0 and no Pfam comparison result, the transcript was identified without coding potential and considered to be potential lncRNAs. 

### 2.4. Bioinformatic Analysis

The expression level of lncRNAs was measured with FPKM score, and both Fold Change ≥ 2 and false discovery rate (FDR) < 0.01 served as screening standards of differentially expressed transcripts. The prediction of target genes is based on distance (less than 10 kb) between lncRNAs and probable target genes using genome annotation. Gene Ontology (GO) enrichment analysis of target genes was performed against the Gene Ontology Consortium database (http://geneontology.org/).

### 2.5. RNA Extraction and Reverse Transcription

Total RNA was extracted from leaf, root, sheath, flower and callus using Trizol (Invitrogen). The samples of pistil, seeds at 3, 7, 15 DAP were homogenized in sodium dodecyl sulfate (SDS) RNA-extraction buffer (5 mM EDTA, 150 mM LiCl, 50 mM Tris-HCl, pH 8.0, 1% SDS) and initially extracted with phenol-chloroform (1:1) before being extracted with Trizol as other tissues [16]. The extracted RNA was treated with RNase-free DNase I (Takara, Beijing, China) at 37 °C for 30 min and further qualified with agarose gel electrophoresis to ensure no genomic DNA contamination before reverse transcription. Corresponding 2 μg RNA was reversed transcribed with random primers using a First Strand cDNA Synthesis Kit ReverTra Ace^-α^ (Toyobo, Osaka, Japan) according to the manufacturer’s instructions.

### 2.6. Semi-Quantitative PCR and qRT-PCR Analysis

For semi-quantitative PCR, equal amount of cDNA from different tissues was amplified by specific reverse transcription (RT) primers then electrophoresed in 1% agarose gel. The amounts of transcripts were quantified against levels of transcripts from *Actin* (*LOC_Os03g61970*). Quantitative real-time PCR (qRT-PCR) was performed on a CFX96 real-time system (Bio-rad, Hercules, CA, USA) using Hieff UNICON^®^ Power qPCR SYBR Green Master Mix (Yeasen, Shanghai, China) with three biological replicates [17]. The specificity of amplified products was inspected by melting curves, and the expression level was obtained by calculating the 2^-ΔΔCT^ values and normalized to *Ubiquitin* gene (*LOC_Os03g13170*) [18]. The relevant primers used in this study were listed in Appendix A.

### 2.7. RNA Interference Vector Construction, Plant Transformation and Phenotype Characterization

To confirm the functions of the identified lncRNA in rice seed development, 203 bp region of *TCONS_00023703* was amplified from NIP genomic DNA and inserted into the RNAi destination vector pANDA by Gateway^®^ Technology system (pENTR^TM^/TOPO^®^ Cloning Kit, Gateway^®^ LR Clonase^®^ II Enzyme mix) according to manufacturer’s instructions (Invitrogen). The recombinant vector was introduced into NIP to generate knock-down mutants via Agrobacterium tumefaciens-mediated transformation [13]. At least 100 mature filled grains were used to measure grain length, grain width and 1000-grain weight using an SC-G Automatic test analyzer (Wseen, Hangzhou, China). Relevant primers are listed in Appendix A.

## 3. Results

### 3.1. Genome-Wide Identification of Long Non-Coding RNAs in Rice Developing Seeds

To identify long non-coding transcripts during rice seed development, we performed paired-end RNA-seq on rice mature pistils before pollination (0 DAP) and seeds at 3, 7 days after pollination (3 DAP and 7 DAP). Rice seed development starts from the pistils before pollination, further undergoes double fertilization, rapid proliferation and differentiation of the embryo, then the endosperm cells gradually differentiate into aleuronic cells and starch storage cells and finally to mature seeds. Pistils before pollination and seeds at around 3 and 7 DAP represent the three critical time points before fertilization, rapid division and differentiation of rice seed, respectively [19]. The experiment finally yielded 8.03 × 10^7^, 6.59 × 10^7^, 9.06 × 10^7^ clean reads in 0, 3, 7 DAP datasets, respectively (Appendix A). As shown in Figure 1a, all the sequenced data are of high-quality scores and are suitable for further analysis (Figure 1a). By aligning the sequences to the rice genome IRGSP-1.0 [20], we obtained 19.35 × 10^7^ mapped reads (Appendix A). The quality of the mapped data was further evaluated by testing the (1) randomness of mRNA fragments; (2) length of inserted fragments; (3) degree of saturation of RNA-seq datasets. Figure 1b–d showed normal inserted fragments size (with a peak of about 150 bp), high randomness of RNA fragments and saturated number of detected genes (FPKM score ≥1 was identified as expressed gene), indicating that the transcriptomic library covered most of the genome and was suitable for the identification of lncRNAs.

Three criteria were used for lncRNAs identification in the current study. Firstly, the known coding mRNAs (transcripts and their splices) were filtered out from the mapped reads. Secondly, largely expressed transcripts with length ≥200 bp and FPKM ≥1.5 (calculated using Cufflinks) were selected. In this step, 45,078 potential lncRNA transcripts were identified. Thirdly, these sequences were further processed using algorithms CPC, CNCI and Pfam to assure non-existence of protein-coding domains (Appendix A). At last, we identified a total of 540 highly reliable lncRNAs from rice developing seeds (Figure 2, Appendix A).

To study the basic characteristics of lncRNAs, we compared the lncRNAs with mRNAs (Figure 3). The transcript length of lncRNAs (with peak at 600 nucleotides, ~66% are found within 200–1000 nucleotides (nt), with a mean length of 1147.2 nt) was generally shorter than protein-coding transcripts (with peak at ≥3000 nucleotides, ~ 13.3% are found within 200–1000 nucleotides, with a mean length of 2788.7 nt) (Figure 3a). About 74.8% lncRNAs had between one to fourteen exons with the highest number between one and three (mean of 2.5 and a median of 2), while there are more than 30 exons in the protein-coding transcripts, and the mean of exon number was 8.8 (median of 7) (Figure 3b). It could be seen from Figure 3c that almost 64% of lncRNAs are with (less than 100 residues) or without open reading frame (ORF), and a mean ORF length of 107.8. In contrast, about 85.7% of mRNAs had longer ORF (more than 100 residues) and a mean ORF length of 241.9. Additionally, in terms of FPKM, the transcriptional abundance of lncRNAs were significantly lower than that of mRNAs (Figure 3d).

### 3.2. The Identification and Verification of Differentially Expressed lncRNAs (DELncs)

The FPKM score (Appendix A) was used to evaluate the transcriptional abundance of lncRNAs. DELncs were defined as those lncRNAs with Fold Change ≥ 2 and FDR < 0.01. Out of the 540 lncRNAs, 482 DELncs were differentially expressed among the three seed developing stages (Appendix A) (Table 1), and the hierarchy clustered as shown in Figure 4a. To verify the DELnc pattern, 15 lncRNAs were selected for semi-quantitative PCR analysis in various tissues and developmental stages (Figure 4b,c). As a result, we found that the expression patterns of most of the DELncs (*TCONS_00020279*, *TCONS_00082068*, *TCONS_00071413*, *TCONS_00037941*, *TCONS_00040695*, *TCONS_00020143*) were almost identical to the transcriptomic data (Figure 4b,c). Other lncRNAs were also highly expressed during seed development (*TCONS_00059921*, *TCONS_00028756*, *TCONS_00020276*, *TCONS_00020277*) and even expressed specifically in rice seeds (*TCONS_00069814*, *TCONS_00095563*, *TCONS_00027626*, *TCONS_00095152*) (Figure 4b,c). 

As lncRNA regulates target genes through cis-acting, we attempted to identify genes within 10 kb upstream and downstream of the 421 DELncs, which are likely to be the potential targets (Appendix A). The potential target genes encode proteins with various functions, such as transposon and retrotransposon proteins, arginine/serine-rich protein, transcriptional factors, cell number regulator, disease resistance protein, formin-like protein, Pentatricopeptide repeat-containing protein, and vegetative cell wall proteins. Furthermore, the target genes of DELncs in groups 0 DAP vs. 3 DAP, 0 DAP vs. 7 DAP and 3 DAP vs. 7 DAP were analyzed with GO enrichment analysis. The numbers of target genes classified by GO were 507, 527, 510 in comparison groups 0 DAP vs. 3 DAP, 0 DAP vs. 7 DAP and 3 DAP vs. 7 DAP respectively. GO analysis in these three groups showed that these target genes were actively affected in most of the categories in the cellular component, molecular function and biological process (Figure 5). The most highly represented groups were involved in cellular processes, metabolic processes, catalytic activities, organelles and cell parts in these three groups. These results suggested that these differentially expressed lncRNAs may participate in seed development by affecting various pathways.

### 3.3. RNA Interference Analysis Reveals the lncRNA Participating in Seed Development

The seed-specific expression pattern of the lncRNAs intrigued us to test their biological functions in seed development. We generated RNAi lines of three seed-specific lncRNAs, however, lncRNA *TCONS_00023703* attracted our particular interest due to the obvious phenotype in seeds. The expression patterns of *TCONS_00023703* showed that it was highly expressed in developed seeds, especially in the early stages of seed development (3 and 7 DAP), but hardly expressed in other tissues (Figure 6a). For three generations (T_1_–T_3_), the RNAi mutants of *TCONS_00023703* all showed significantly decreased expression levels and consistent phenotypes including reduced grain length, width and 1000-grain-weight (Figure 6b–e). Target gene prediction analysis of *TCONS_00023703* suggested that *LOC_Os11g17480* is a potential target. This gene is located within the 10 kb downstream region of *TCONS_00023703* and is annotated as a dienelactone hydrolase belonging to the Alpha/beta-Hydrolases subfamily (https://www.uniprot.org/). However, its detailed function in seed development has not been reported. Moreover, we examined the transcription level of several reported grain-size-related genes in the wild-type (WT) and *TCONS_00023703* RNAi line using qRT-PCR and found that the expression levels of *GS2*, *GW2*, *GS5* were significantly up-regulated and *OsSPL13* was significantly down-regulated in *TCONS_00023703* compared with WT (Figure 7). In addition, positive regulators *GS3* and *qTGW3* were also slightly up-regulated in *TCONS_00023703* RNAi plant. These results indicated that lncRNA *TCONS_00023703* functions as a positive regulator in grain size and weight.

## 4. Discussion

Long non-coding RNAs are ubiquitous in eukaryotes, and their functions have been intensively studied in humans, especially in diseases such as cancer [21,22]. Although plant lncRNA research is still in its infancy, the rapid development and application of transcriptome technology has led to the identification of a large number of lncRNAs in plants [23,24,25,26]. Many lncRNAs have been found to be involved in the regulation of important agronomic traits [11], cadmium stress response in root [27], female reproduction [28], male reproduction [23] and other stress responses [29] in rice. In addition, 578 lncRNAs related to high-temperature-induced grain chalkiness development were identified in rice spikelets harvested at 10 DAP exposed to high-temperature stress, and expression levels of some of them were verified by RT-PCR [30]. However, there are no reports regarding the identification of lncRNAs in rice developing seeds. In our study, three key time points (0, 3, 7 DAP) for early seed development were obtained as samples for transcriptome sequencing, and a total of 540 lncRNAs were screened after bioinformatics analysis. To confirm sequencing quality, fifteen lncRNAs were randomly selected for semi-quantity PCR validation (Figure 4b,c). The results showed that the expression levels of the lncRNAs are consistent with the transcriptomic data. All of them showed the highest expression levels in seeds or even in a seed-specific manner (Figure 4b,c). Moreover, a total of 482 lncRNAs are significantly differentially expressed in the three samples, suggesting that lncRNAs may be an important regulator involved in early seed development (Table 1). Despite the tremendous progress in the identification of lncRNAs, few of them are functionally characterized thus far. In rice, a lncRNA *LAIR* was identified and found to be regulated in grain yield by changing its expression level [31]. Plants overexpressing *LAIR* exhibited larger primary panicles and more panicles per plant while silencing *LAIR* decreased grain yield [31]. Zhanget al [12] performed RNA sequencing and identified lncRNAs that are involved in the reproduction of rice, and studied the function of one lncRNA *XLOC_057324* by Tos17 insertion. This lncRNA was specifically expressed in young panicles and the insertion mutant exhibited early flowering and decreased fertility compared with wild-type plants. In this study, *TCONS_00023703*, one of the differentially expressed lncRNAs screened with our bioinformatics analysis, had its expression pattern verified, and also its function was described using RNAi analysis in rice. The results revealed that *TCONS_00023703* was highly expressed in developing seeds and its three generations of mutant plants showed a significant decrease in grain length and 1000-grain weight (Figure 6a,e). The above results not only verified the transcriptome results but also established that lncRNAs can positively regulate rice grain length, which lays the foundation for further study in regulation mechanism of lncRNAs. 

As important yield traits in rice, many genes controlling grain size and weight have been reported [32,33,34]. *GW2*, *GS3* and *qTGW3* negatively regulated grain size and weight in rice [35,36]. Consistent with this, they were up-regulated in *TCONS_00023703* compared with WT (Figure 7) While *OsSPL13*, a positive regulator, was significantly down-regulated in *TCONS_00023703* RNAi plant. These implied the reduced grain size in *TCONS_00023703* may be related to the altered expression of these genes (Figure 7). However, *GS2* and *GS5* were reported to have positively regulated grain size in rice [37,38,39,40,41]. Contrary to our understanding, these two genes were also up-regulated in *TCONS_00023703* RNAi plant (Figure 7). However, little is known about regulatory mechanisms of lncRNAs, its accurate regulation mechanism remains to be further studied.

## 5. Conclusions

In conclusion, a pair-end RNA sequencing of rice pistil, 3 DAP seeds and 7 DAP seeds identified a total of 540 lncRNAs, of which 482 were differentially expressed among the three developing stages. The transcriptomic data proved to be highly reliable by semi-quantity PCR analysis of 15 selected DELncs in various tissues. Knock-down of a DELnc *TCONS_00023703* in rice resulted in smaller grain size and weight when compared with the WT.

## Figures and Tables

**Figure 1 genes-11-00243-f001:**
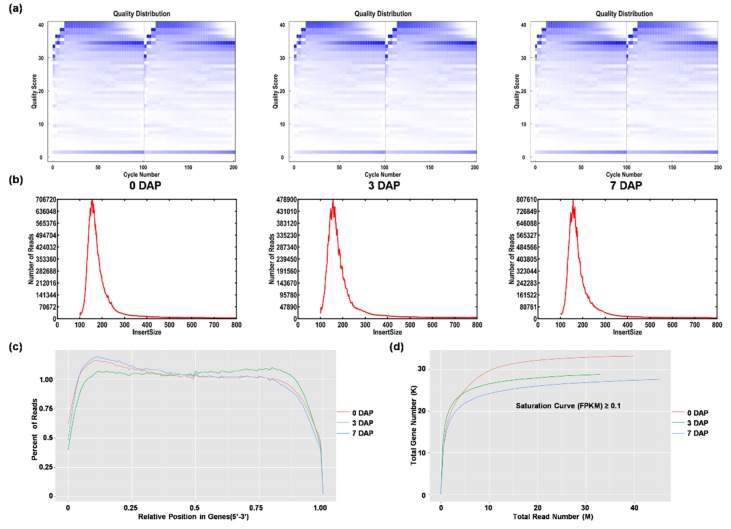
Quality score distribution of each raw data and quality assessment of transcriptome library for 0, 3, 7 days after pollination (DAP) seeds. (**a**) Quality score distribution of each raw data. The abscissa is the number of sequencing and the ordinate is the quality score (10, 20, 30, 40 represent probability of base identification error is 1/10, 1/100, 1/1000, 1/10000, respectively.). The depth of blue color indicates the base-specific gravity, and the darker the color, the larger the proportion of the bases. (**b**) Inserted fragments size of each raw data. (**c**) Randomness test of RNA fragments. The abscissa is the normalized messenger RNA (mRNA) position, that is, each mRNA is divided into 100 intervals according to the length, and then the number and proportion of mapped reads in each interval. (**d**) Saturation test of transcriptome sequencing data.

**Figure 2 genes-11-00243-f002:**
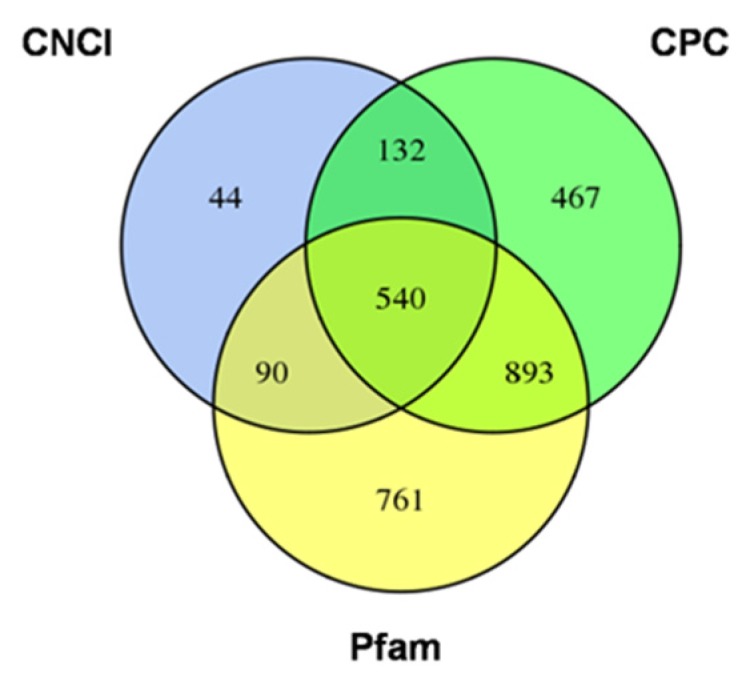
The Venn diagram of the potential long non-coding RNAs (lncRNA) transcripts screened by coding potential analysis methods including CNCI (Coding-non-coding index), CPC (Coding potential calculator), Pfam (Protein family) database analyses.

**Figure 3 genes-11-00243-f003:**
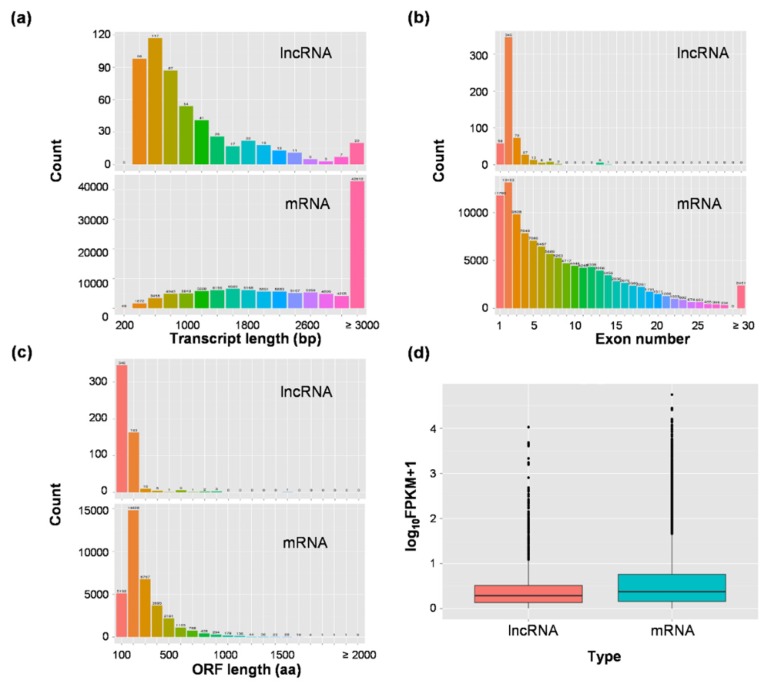
Comparative analysis of mRNAs and identified lncRNAs. (**a**) Comparison of transcript lengths of lncRNAs and mRNAs. (**b**) Comparison of the number of exons of mRNAs and lncRNAs. (**c**) Comparison of the length of open reading frame (ORF) of mRNAs and lncRNAs. (**d**) Comparison of expression levels of mRNAs and lncRNAs. The ordinate indicates the logarithm of the sample expression amount FPKM (fragments per kilobase of transcript per million fragments mapped).

**Figure 4 genes-11-00243-f004:**
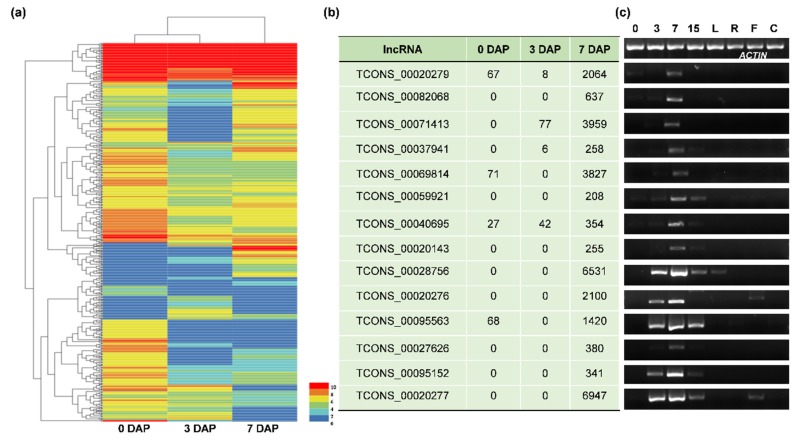
Differentially expressed lncRNAs. (**a**) A cluster heatmap illustrating the differentially expressed lncRNAs of RNA sequencing results in 0, 3, 7 DAP seeds. Colors in the map represent the level of expression in the sample from the FPKM values (log_2_FPKM+1). (**b**) Partial transcriptome results of identified lncRNAs expression levels. The values are FPKM scores. (**c**) Validation of expression patterns of lncRNAs corresponding to the left values in (**b**) using semi-quantitative PCR. *ACTIN* was used as an internal loading control. 0, 3, 7, 15 represent pistil, seeds at 3, 7, 15 days after pollination, respectively; L: Leaf; R: Root; F: Flower; C: Callus.

**Figure 5 genes-11-00243-f005:**
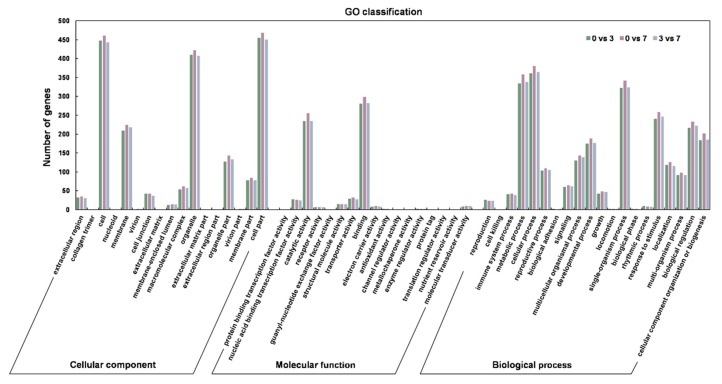
Comparison of gene ontology (GO) classification for the predicted target genes of the differentially expressed lncRNAs in groups 0 DAP vs. 3 DAP, 0 DAP vs. 7 DAP, 3 DAP vs. 7 DAP.

**Figure 6 genes-11-00243-f006:**
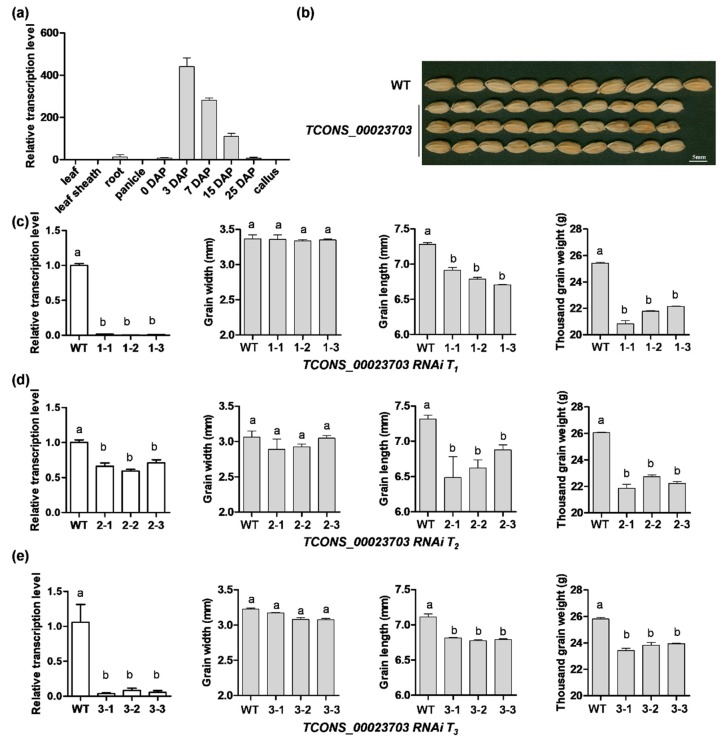
Functional analysis of lncRNA TCONS_00023703. (**a**) The expression pattern of TCONS_00023703 using quantitative real-time PCR. Samples from different organs and stages were of Nipponbare. (**b**) Grain size in *TCONS_00023703* knock-down mutant plants compared with wild-type (WT) (Nipponbare). Scale bar = 2 mm. (**c**–**e**) The three generations of knock-down mutants showed significantly down-regulated expression in *TCONS_00023703*, a slight decrease in grain width, and a significant decrease in grain length and thousand-grain weight. Data was determined by student’s *t*-test to generated *p*-values. Different letters indicate *p* < 0.01.

**Figure 7 genes-11-00243-f007:**
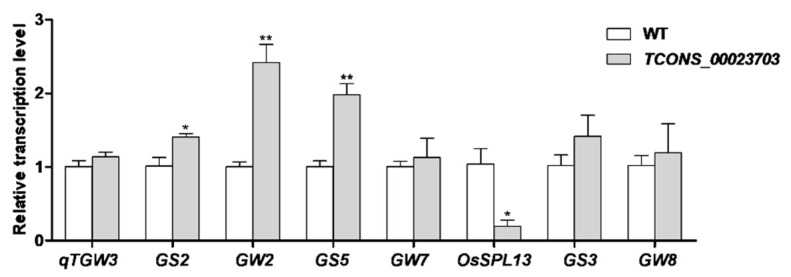
Quantitative RT-PCR analysis of genes that determine grain size. The expression analysis was conducted using seeds after pollination at three days in WT (wild type) and *TCONS_00023703* knock-down mutant. * and ** represent a significant difference between wild type and mutant at the 0.05 and 0.01 level, respectively.

**Table 1 genes-11-00243-t001:** Statistical table of differentially expressed lncRNAs.

DEG Set	DELncs Number	Up-Regulated	Down-Regulated
0 DAP *vs* 3 DAP	447	190	257
0 DAP *vs* 7 DAP	454	218	236
3 DAP *vs* 7 DAP	440	246	194

Differentially expressed genes (DEG) set: Pairwise comparison of samples of differentially expressed lncRNA; DELncs number: Number of differentially expressed lncRNAs; Up-regulated: Number of up-regulated lncRNAs; Down-regulated: Number of down-regulated lncRNAs.

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
