# Peer review of "Genome-Wide Identification of lncRNAs During Rice Seed Development"

_genes, 2020, doi:10.3390/genes11030243_

Round 1

Reviewer 1 Report

The authors carried out paired-end RNA sequencing using rice samples at different developmental status. Differential expressed IncRNAs were analysed. One of the highly expressed IncRNAs was studied by RNA interference and is important for agronomic traits. However, the following questions need to be addressed before publication.

What is the criterion used to select which IncRNA is used for RNAi experiment?

Could the authors address the significance of the identification of DELnc TCONS_00023703 in details?

What is the expression pattern of LOC_Os11g17480 as a potential target? Does the expression corelated to TCONS_00023703 during seed development?

What is the distribution of IncRNAs on rice chromosome? Are they located in a cluster?

Figure 4, a; Could they authors add annotation to each row of the heatmap?

Author Response

Responds to the reviewer’s comments:

Comments and Suggestions for Authors

The authors carried out paired-end RNA sequencing using rice samples at different developmental status. Differential expressed IncRNAs were analysed. One of the highly expressed IncRNAs was studied by RNA interference and is important for agronomic traits. However, the following questions need to be addressed before publication.

  1. What is the criterion used to select which IncRNA is used for RNAi experiment?

Response: we selected lncRNAs with seed-specific expression pattern and have high expression levels both in RNA sequencing and Semi-quantitative PCR analysis.

  1. Could the authors address the significance of the identification of DELnc TCONS_00023703 in details?

Response: rice is an important food crop, and its yield affects global human food safety. It is of great significance to study genes and lncRNAs related to rice yield. Because TCONS_00023703 is specifically expressed in seeds and the TCONS_00023703 RNAi plants have reduced grain size, it indicates that it can regulate the process of seed development and may affect rice yield. Therefore, the identification of lncRNA TCONS_00023703 lays the foundation for further research on its regulatory mechanism in yield traits.

  1. What is the expression pattern of LOC_Os11g17480 as a potential target? Does the expression corelated to TCONS_00023703 during seed development?

Response: according to Rice Expression Database (http://expression.ic4r.org/index), LOC_Os11g17480 is most expressed in the roots, followed by panicles and seeds. In addition, according to our transcriptome data (data not shown in this paper), this gene has the highest expression in the three days of seed samples in seeds at 3 DAP (FPKM are 0.046852, 0.355119. 0.224855 in seeds at 0, 3, 7 DAP, respectively). This is somehow different from the expression pattern of TCONS_00023703, but its expression pattern is consistent at different stages of seed development.

  1. What is the distribution of IncRNAs on rice chromosome? Are they located in a cluster?

Response: as shown in Supplementary Table 8, the identified lncRNAs in our study distributed on 12 chromosomes in rice, and there is no obvious cluster distribution.

  1. Figure 4, a; Could they authors add annotation to each row of the heatmap?

Response: each row of the heatmap in Figure 4a represents a lncRNA. Due to too large data, it cannot be shown in Figure 4a, and its raw data of RNA-seq will be uploaded to NCBI database. The authors sincerely appreciate your efforts for this paper.

Reviewer 2 Report

In this study, the authors have conducted RNA-seq analysis in means to detect lncRNAs during rice seed development. Their objective was to identify lncRNAs that are related with seed development and had roles in gene regulation. The description of the analysis workflow was well designed and clearly described and following experiments were well aligned with the provided results. Some comments are given as below.

Major comments
1. Instead of choosing 15 lncRNAs randomly, it may have been a better choice to select them in respect of cis-acting lncRNAs based on their position or expression levels.
2. Out of the 540 lncRNAs detected, the number of DElncRNAs seems too high, which was 482. The p value may need to be further corrected.
3. The authors have specifically focused on the TCONS_00023703 lncRNA. Is this lncRNA studied in other plants than rice? Search across different plants of this lncRNA may help shed some further explanation.
4. The english was well written with few typos observed.
5. What is the unit of expression level in Fig 4(b)? lncRNAs with excessive expression level tend to be false positive and should be discarded from the analysis, such as TCONS_00073105 in Fig4(b).
6.

Minor comments
1. Some minor typos should be corrected as below
L283 - to regulated grain... -> to be regulated grain
L286 - The author initials of the reference should be discarded

Author Response

Responds to the reviewer’s comments:

Comments and Suggestions for Authors

In this study, the authors have conducted RNA-seq analysis in means to detect lncRNAs during rice seed development. Their objective was to identify lncRNAs that are related with seed development and had roles in gene regulation. The description of the analysis workflow was well designed and clearly described and following experiments were well aligned with the provided results. Some comments are given as below.

Major comments

   1. Instead of choosing 15 lncRNAs randomly, it may have been a better choice to select them in respect of cis-               acting lncRNAs based on their position or expression levels.

Response: Nice suggestion! We actually selected lncRNAs with very different expression levels in the three developmental stages of the seeds based on the expression levels in transcriptome data (Supplementary Table 7) for subsequent research.

  1. Out of the 540 lncRNAs detected, the number of DElncRNAs seems too high, which was 482. The p value may need to be further corrected.

Response: In this study, DELncs were defined as those lncRNAs with Fold Change ≥ 2 and FDR (False Discovery Rate) < 0.01, which is a quite stringent criteria and has been widely used in several reported cases (Leszczynska et al, 2010; Hao et al, 2015; Zhang et al, 2016). We believe 482 DElncRNAs is still an acceptable number for the followed analysis in this study .

Leszczynska, A. , Hoser, G. , Kotlinski, M. , Prymakowska-Bosak, M. , Szkopinska, A. , & Burzynska, B. . (2010). The effect of different statins on gene expression profile of human hepatoma cells. New Biotechnology, 27(supp-S1), 0-0.

Hao, L. , Beiqin, Y. , Jianfang, L. , Liping, S. , Min, Y. , & Jun, Z. , et al. (2015). Characterization of differentially expressed genes involved in pathways associated with gastric cancer. PLOS ONE, 10(4), e0125013.

Zhang, H. , Hu, W. , Hao, J. , Lv, S. , Wang, C. , & Tong, W. , et al. (2016). Genome-wide identification and functional prediction of novel and fungi-responsive lincrnas in triticum aestivum. BMC Genomics, 17(1), 238.

  1. The authors have specifically focused on the TCONS_00023703 lncRNA. Is this lncRNA studied in other plants than rice? Search across different plants of this lncRNA may help shed some further explanation.

Response: to the best of our knowledge, TCONS_00023703 has not been reported in any other plants.

  1. The english was well written with few typos observed.

Response: corrections have been made to improve the readability.

  1. What is the unit of expression level in Fig 4(b)? lncRNAs with excessive expression level tend to be false positive and should be discarded from the analysis, such as TCONS_00073105 in Fig4(b).

Response: the FPKM values in Figure 4b represent relative expression levels which do not have a unit. TCONS_00073105 in Figure 4b is excluded.

Minor comments
1. Some minor typos should be corrected as below
L283 - to regulated grain... -> to be regulated grain
L286 - The author initials of the reference should be discarded.

Response: corrected. The authors sincerely appreciate your efforts for this paper.